# SELF-SUPERVISION MEETS BOOTSTRAP ESTIMATION: NEW PARADIGM FOR UNSUPERVISED RECONSTRUCTION WITH UNCERTAINTY QUANTIFICATION

## ABSTRACT

Deep learning-based self-supervised reconstruction (SSR) plays a vital role in diverse domains, including unsupervisedly reconstructing magnetic resonance imaging (MRI). Current powerful methodologies for self-supervised MRI reconstruction usually rely on capturing the relationships between different views or transformations of the same data such as serving as inputs and labels respectively, which show notable influence from analogous approaches in computer vision. Although yielding somewhat promising results, their designs are often heuristic without deep insights into reconstructed object characteristics, and the analytical and mathematical principles of such methods are not expressive. This paper addresses these issues with a novel SSR paradigm, *BootRec*, that not only provides an explanation for self-supervised reconstruction but also facilitates the development of downstream algorithms. Self-supervised MRI reconstruction is modeled as error-oriented parameter estimation - Bootstrap estimation for SSR (BootRec). In BootRec, we demonstrate the mathematical equivalence between bootstrapping in a sample set and the commonly used re-undersampling operation for SSR. This insight is further incorporated into designing models to estimate errors of MRI SSR results without accessing labeled data. The estimation can further serve as the loss function for unsupervisedly training the models. Experiments show that our new paradigm BootRec enables advanced MRI reconstruction performance against other zero-shot methods. The code is available at https://github.com/user19781945/rep10825984.

## 1 INTRODUCTION

Magnetic resonance imaging (MRI) reconstruction receives continuous attention for its significance in medical imaging and challenges in often unsupervised settings due to costly labeling and obtaining ground truth. MRI reconstruction inherently requires a lengthy step of repeatedly collecting measurements in the frequency domain to fill the k-space before recovering the spatial signals using inverse Fourier transform (IFT). The advancements in techniques such as parallel imaging for acquiring signals and compressed sensing (CS) for reconstruction have provided approaches to reduce imaging time. Specifically, CS makes it possible to acquire fewer measurements than the Nyquist rate while reducing the aliasing artifacts (Donoho, 2006; Lustig et al., 2008).

The introduction of deep neural networks for deep learning (DL) to CS-MRI has also led to breakthroughs in a higher acceleration ratio and better reconstruction quality in MRI reconstruction (Chen et al., 2022; Wang et al., 2021; Lin & Heckel, 2022; Fabian et al., 2021). However, these DL methods, though powerful, have several challenges in further applications. The first problem is that supervised DL training demands numerous labeled training data. In the situation of MRI reconstruction, it means that enough fully sampled images must be provided, which is impossible in many situations. Another important shortcut is the black-box nature of DL models, making the reconstruction lack explanation and uncertainty estimation. Hence, it is hard to evaluate the risk in real-world medical practice when doctors need to make critical decisions according to the images (Edupuganti et al., 2020).

We propose a new paradigm of Bootstrap estimation for self-supervised reconstruction (BootRec) of MRI. BootRec models MRI SSR as a parameter estimation problem, and applies Bootstrap estimation to quantify the errors. The learning target is then shifted to minimize the estimated mean squared error (MSE) between the reconstructed fully sampled images and the unknown ground truth. Summary of different pipelines and the insights of our modeling are in Figure 1.

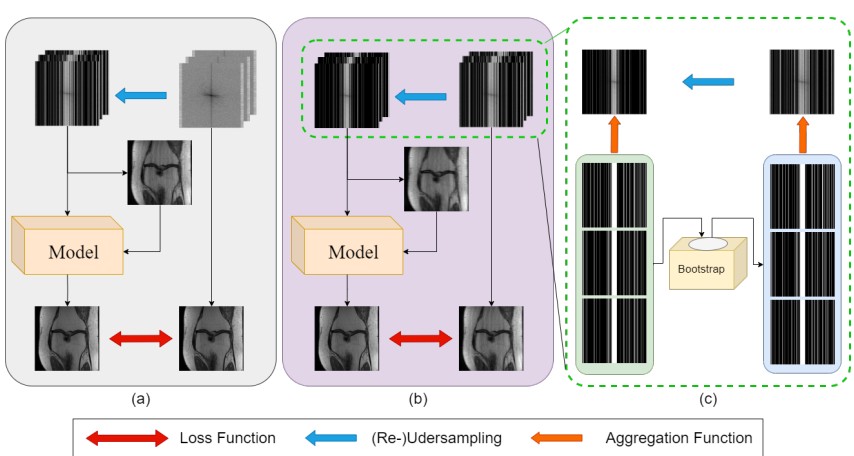

Figure 1: Demonstration of different pipelines of DL-based reconstruction models. All self-supervised methods incorporate some kind of **re-undersampling**. (a) Supervised training with paired fully sampled images as labels. (b) self-supervision via data undersampling pipeline. (c) Insights of modeling re-undersampling as Bootstrap. Only masks of virtual sample sets are plotted for simplification.

The main contributions of our BootRec paradigm are summarized as follows: (1) We construct a new framework that models MRI SSR as a parameter estimation problem. (2) We demonstrate the equivalence between Bootstrap sampling and re-undersampling in certain conditions. (3) We propose using Bootstrap MSE estimation as uncertainty quantification for SSR. (4) We propose new algorithms to train self-supervised models and achieve advanced results.

The notations used in the paper are summarized in Appendix A for reference.

## 2 BACKGROUND & RELATED WORK

### 2.1 DEEP-LEARNING-BASED RECONSTRUCTION FOR MRI

The imaging process of parallel CS-MRI in one coil can be formulated as Equation 1, where $y$ represents the acquired k-space data, $x$ is the spatial anatomy data, $r$ is the noise, $\mathcal{F}$ is the Fourier transform, $U$ is the 0-1 valued matrix indicating the sampling points in k-space (which is called measurement matrix in CS), and $C$ is the coil sensitivity. Note that $x$, $y$ and $r$ should be multi-dimensional values. The notions in the paper are summarized in Appendix A. For simplification and to be consistent with other references, we represent them as flattened vectors.

$$y = U\mathcal{F}Cx + r \tag{1}$$

For simplification in the later analysis and without influencing the conclusion, we will skip the combination of multiple coils and ignore the noise $r$ as it is usually modeled as Gaussian noise with mean of zero. We also ignore the coil sensitivity or merge them into $x$, thus obtaining a simplified equation of CS-MRI:

$$y = U\mathcal{F}x \tag{2}$$

Given the acquired $y$, the reconstruction is built as a reverse problem to recover $x$ using some reconstruction model. Traditionally, the reconstructor is iterative based on CS theory, while in deep-learning-based methods, the model can be a neural network parameterized by $\theta$(Chen et al., 2022; Yang et al., 2016). We represent any reconstruction model as Equation 3.

$$\hat{x} = f(y, U) \tag{3}$$

## 2.2 Self-Supervised Training of Reconstruction Models

Early trials of unsupervised training of reconstruction models implement dictionary learning and other classical algorithms in CS (Majumdar, 2018; Singhal & Majumdar, 2020). Other methods include leveraging unpaired fully-sampled data (Oh et al., 2020; Chung et al., 2021; Korkmaz et al., 2022) and Deep Image Prior (DIP) (Ulyanov et al., 2018). Benefiting from the success of self-supervised methods in computer vision like contrastive learning (Chen et al., 2020) and masked Autoencoder (He et al., 2022), self-supervised training in MRI reconstruction has made progress in recent years and surpassed other methods (Zhou et al., 2022; Yaman et al., 2020; Zou et al., 2022; Wang et al., 2022b).

The basic pipeline of self-supervised model is in Figure 1. As shown in Equation 4t, reconstruction is conducted on the re-undersampled measurements $y$, and the basic form loss of is $\mathcal{L}(\hat{x}^R, \hat{x})$. The key points are the design of re-undersampling masks $\boldsymbol{U}^R$ and loss functions. In different models (Wang et al., 2022b; Yaman, 2022), different kinds of sampling methods (uniform, Gaussian, etc.) and ratios of re-undersampling are proposed and evaluated. The loss function of the self-supervision mainly comes from the undersampled k-space not being selected in re-undersampling, which can be defined in the frequency or spatial domain (Jafari et al., 2021; Senouf et al., 2019), with a wide range of choices from imaging processing.

$$\hat{\boldsymbol{x}^R} = f(\boldsymbol{U}^R\boldsymbol{y}, \boldsymbol{U}) \tag{4}$$

Generally speaking, the explorations of effective self-supervised algorithms for CS-MRI reconstruction are heuristic. Instead, Bootrec will try to provide a methodology and explanation for this field.

## 2.3 Uncertainty Quantification of MRI Reconstruction

DL models show impressive advantages in many fields with a major concern about their result's reliability, such as the hallucination of large language models (OpenAI, 2023). In MRI reconstruction, a concern is that DL models may "imagine" the anatomies and mislead the diagnosis. Uncertainty Quantification (UQ) can ameliorate the problem by providing "confidence level" of the results, enabling decision-makers aware of the risk of unauthentic imaging (Gawlikowski et al., 2021), and doctors can choose to conduct further examinations for results of high uncertainty.

Derived from its origin, uncertainty of reconstruction can be divided into two categories (Kendall & Gal, 2017), aleatoric uncertainty stemming from the ill-posedness of the problem and epistemic uncertainty from the uncertainty of model parameters. The notion of uncertainty is also to be clarified. In the field of image tasks, the variance of the result is widely used, and other choices include quantiles and entropies(Angelopoulos et al., 2022). In MRI reconstruction and other image regression tasks, the residual error of the prediction also made notable progresses(Wang et al., 2022a).

Uncertainty quantification has been considered in the community of computational imaging. In the field of MRI reconstruction, Edupuganti et al. (2020) leverages the variational Autoencoder (VAE) to convert the deterministic result to be probabilistic. Schlemper et al. (2018b) and Ekmekci & Cetin (2022) builds a Bayesian neural network (BNN) and models the inherent uncertainty with a Gaussian distribution. A main limitation of existing methods is that supervised training is needed for the quantification so they cannot be applied to unsupervised models.

We find Bootstrap estimated MSE can be viewed as UQ to some extent, which models the aleatoric uncertainty from (re)-undersampling well. Further experiments are conducted to assess the quantification.

## 3    Modeling Reconstruction as Parameter Estimation

The BootRec framework consists of the following modules: (1) *aggregation function* for preprocessing and wrap reconstruction model as parameter estimator; (2) *a virtual sample set* as a mathematical tool to map a single observation to a sample set; (3) *pseudo resampling trick* to map Bootstrap sampling of a virtual sample set to re-undersampling of measurement; and (4) *training algorithms* for the specific loss function based on bootstrapping. These are detailed below.

## 3.1 Distribution of MRI Acquisition Observation

Firstly, we assume the sampling mask $U$ obeys a multivariate Bernoulli distribution where each variable is independent, as Equation 5. We also make a constraint that all positions keep the possibility to be sampled, that is, $P_{U_i} > 0$ for any given position $i$.

$$\mathbf{U} \sim \mathcal{B}(\boldsymbol{U}; 1, \boldsymbol{P_U}) \tag{5}$$

BootRec initially operates by training a separated model for each data point (zero-shot reconstruction(Yaman, 2022)) and we'll discuss more general situation in Section 5.3. In the zero-shot case, the target of the $i_{th}$ reconstruction is fixed as $x^{(i)}$, so we directly use $\boldsymbol{x}$ as $x^{(i)}$ in the following derivation.

With former Equation 2, the randomness from the mask is introduced so the sampled k-space data can also be viewed as random variables. Usually, the sampling mask and the acquired k-space data are provided and processed simultaneously in CS, so we define the observation as $\boldsymbol{s} = (\boldsymbol{y}, \boldsymbol{U})$ for convenience. The distribution of $\mathbf{s}$ can be fully parameterized by $\boldsymbol{x}$ and $\boldsymbol{P_U}$, written as $p(\mathbf{s}; \boldsymbol{x}, \boldsymbol{P_U})$. The reconstruction task in Equation 3 can then be viewed as estimating the parameter of $p(\mathbf{s})$ given observations of s.

## 3.2 Estimator Construction with Aggregation Function

To formulate estimators from the reconstruction models, the main distinction is that observations are processed individually and independently without forming a set, as in Equation 3, so we propose *aggregation function* as an adapter.

An aggregation function is a special mapping from observation sets $\{\boldsymbol{s}^{(1)}, \boldsymbol{s}^{(2)}, \dots \boldsymbol{s}^{(n)}\}$ to a single observation $\boldsymbol{s}^* = (\boldsymbol{y}^*, \boldsymbol{U}^*)$ and serves as a component of the estimator. The output can then be directly passed to any existing reconstruction model.

In the situation of $n$ (positive integer) independent observations $\{\boldsymbol{s}^{(1)}, \boldsymbol{s}^{(2)}, \dots \boldsymbol{s}^{(n)}\}$, the aggregation function is defined as Equation 6. Intuitively it takes the average observed value in positions being selected at least once and keeps the other positions zero-valued. A case of $n = 3$ is demonstrated in Figure 2.

$$h(\boldsymbol{s}^{(1)}, \boldsymbol{s}^{(2)}, \dots \boldsymbol{s}^{(n)})_i = (\boldsymbol{U}^*\boldsymbol{y}, \boldsymbol{U}^*)_i = (\boldsymbol{y}^*, \boldsymbol{U}^*)_i \tag{6}$$

$$\boldsymbol{U}_i^* = \left\{ \begin{array}{ll} 1 & (\boldsymbol{U}^{(1)} + \boldsymbol{U}^{(2)} + ...\boldsymbol{U}^{(n)})_i \neq 0 \\ 0 & (\boldsymbol{U}^{(1)} + \boldsymbol{U}^{(2)} + ...\boldsymbol{U}^{(n)})_i = 0 \end{array} \right. \tag{7}$$

After the aggregation function, it is obvious that $\boldsymbol{y}^*$ will still be k-space data in which all the selected measurements and gradients in $\boldsymbol{U}^*$ remain the same as that of the corresponding position in $\boldsymbol{y}$, as a normal masking operation. Also, if only one observation is acquired, the aggregation function will be transparent and will not be adjusted.

Aggregation function can be composed by any reconstruction method to form a new reconstruction method $f_{AF} = f \circ h$. The new function can take the sample set from the distribution of $P(\mathbf{s})$ and serve as an estimator of the parameter $\boldsymbol{x}$ without modifying the reconstruction process defined by $f$.

$$\hat{\boldsymbol{x}} = f_{AF}(\boldsymbol{s}^{(1)}, \boldsymbol{s}^{(2)}, \dots \boldsymbol{s}^{(n)}) \tag{8}$$

## 3.3 Equivalent Sample Distribution

Under the definition of aggregation function, $\boldsymbol{U}^*$ means a "selected at least once" matrix and $\boldsymbol{U}^*$ obeys a multivariate Bernoulli distribution whose distribution parameter can be computed as Equation 10. The parameterized distribution of output observations is then $p(\mathbf{s}^*; \boldsymbol{x}, \boldsymbol{P_{U^*}})$ correspondingly.

$$\mathbf{U}^* \sim \mathcal{B}(\boldsymbol{U}^*; 1, \boldsymbol{P_{U^*}}) \tag{9}$$

$$diag(\boldsymbol{P_{U^*}}) = \boldsymbol{I} - (\boldsymbol{I} - diag(\boldsymbol{P_U}))^n \tag{10}$$

Observing the process of aggregation function, we notice that multiple sample sets may be mapped to the same observation. Given a sample set containing $n$ identically distributed observations

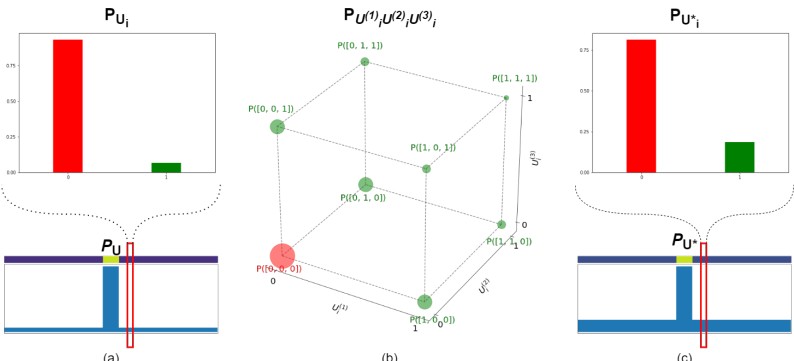

Figure 2: Demonstration of Aggregation Function with 3 observations (only the distribution of masks are presented for simplification). (a)/(c): The probability of being selected in different positions ($\boldsymbol{P}_U$) and the distribution of a specific position being selected or not ($\mathbf{P}_{U_i}$) before/after aggregation function. (b) The distribution of selections in the set of when size $n = 3$, and the color means the corresponding value after the aggregation function.

$\{\boldsymbol{s}^{(j)} | \mathbf{s}^{(j)} \sim p(\mathbf{s}), j = 0, 1, \ldots n\}$, if the result of adding it to the aggregation function satisfy that $h(\boldsymbol{s}^{(1)}, \boldsymbol{s}^{(2)}, \ldots \boldsymbol{s}^{(n)})_i = (\boldsymbol{y}^*, \boldsymbol{U}^*)_i$, we define $p(\mathbf{s})$ the *n-cardinality Equivalent Sample Distribution* of $p(\mathbf{s}^*)$, whose parameter $\boldsymbol{P}_U$ satisfies Equation 10. The two distributions are connected by the aggregation function. In Figure 2, if (c) visualizes the distribution of actual observations, then (a) shows an equivalent sample distribution when $n = 3$. Solving Equation 10 by setting $\boldsymbol{P}_U$ as unknowns, we get Equation 11, which computes the parameters of equivalent sample distribution.

$$diag(\boldsymbol{P}_{U_i}) = \boldsymbol{I} - (\boldsymbol{I} - diag(\boldsymbol{P}_{U^*}))^{1/n} \tag{11}$$

## 4 CONNECTING BOOTSTRAP WITH RE-UNDERSAMPLING

### 4.1 BOOTSTRAP ESTIMATION OF MULTIPLE OBSERVATIONS

For parameter estimation, the inference of the population is performed with the collected sample set of a certain size $n$. However, without reference to the population, the quality of the estimation cannot be computed. Bootstrap method solves the problem by sampling a new sample set of the same size $n$ in the original sample set (with replacement) $m$ times, and using the resampled sets (called Bootstrap sample sets) to form $m$ Bootstrap estimations. The quality of the estimation with the original sample set can then be inferred by assessing the Bootstrap estimation with respect to the original estimation, which is accessible. With much more that can be studied in applying bootstrapping, we here only focus on the non-parameterized Bootstrap method and the Bootstrap estimation of MSE.

In the scale of our modeled reconstruction problem, we can represent the sample set of size $n$ with $\{\boldsymbol{s}^{(1)}, \boldsymbol{s}^{(2)}, \ldots \boldsymbol{s}^{(n)}\}$ and the original estimation as $\hat{\boldsymbol{x}}$. The $k_{th}$ Bootstrap sample set can be represented by $\{\boldsymbol{s}^{B_k(1)}, \boldsymbol{s}^{B_k(2)}, \ldots \boldsymbol{s}^{B_k(n)}\}$, and the corresponding estimation as Equation 12.

$$\hat{\boldsymbol{x}}^{B_k} = f_{AF}(\boldsymbol{s}^{B_k(1)}, \boldsymbol{s}^{B_k(2)}, \ldots \boldsymbol{s}^{B_k(n)}) \tag{12}$$

The MSE of the estimation $\hat{\boldsymbol{x}}$ can be estimated by bootstrapping using Equation 13. For MRI reconstruction, we can see that without reference to the fully sampled image $\boldsymbol{x}$, it is still possible to estimate the MSE of the reconstruction result.

$$\hat{mse}(\hat{\boldsymbol{x}}) = \frac{1}{m} \sum_{k=1}^{k=m} (\hat{\boldsymbol{x}}^{B_k} - \hat{\boldsymbol{x}})^2 \tag{13}$$

## 4.2 VIRTUAL SAMPLE SET AND PSEUDO RESAMPLING TRICK

In the previous section, we show that we can measure the quality of MRI reconstruction using Bootstrap method. However, in real-life applications, it is unrealistic to assume that there will be multiple observations to form a sample set of enough size to perform bootstrapping. In fact, often only one observation may be available in a specific scan. An intuitive method is to randomly generate a sample set with Equation 6 as a constraint or to assign the points to observations in the virtual sample simply uniformly. These methods will lead to high variance in computation with no prior knowledge leveraged. Instead, we propose to get the distribution of the observations by mapping the observation to a virtual sample set derived from the equivalent sample distribution defined in Section 3.3, which generates estimations equally distributed as a direct reconstruction given the single observation.

If the single observation follows distribution $p(\mathbf{s}^*)$, the equivalent sample distribution is then $p(\mathbf{s})$ correspondingly, which forms the prior distribution of the observations in the virtual sample set of corresponding size $n$. However, given a specific observation $\boldsymbol{s}^* = (\boldsymbol{y}^*, \boldsymbol{U}^*)$, the operation of "sampled at least once" is constrained so we should instead model the observations in the virtual sample set as a conditioned distribution, as formalized in Equation 14.

$$P(\mathbf{U}_i^V | \boldsymbol{U}_i^*) = \mathbf{1}_{\mathbf{U}_i^*=1} Pr(\mathbf{U}_i^V = 1 | \boldsymbol{U}_i^* = 1) \tag{14}$$

We can calculate the probabilities with Bayes' theorem and then use approximate values to help implementation, the result is in Equation 15 and details of derivation can be found in Appendix C. The distribution of virtual sample set elements is parameterized as $p(\mathbf{s}^V; \boldsymbol{x}, \boldsymbol{P}_{\boldsymbol{U}^V})$.

$$\boldsymbol{P}_{\boldsymbol{U}_i^V} = Pr(\mathbf{U}_i^V = 1) = \begin{cases} 1 & \boldsymbol{P}_{U_i} = 1 \\ 1/n & \boldsymbol{U}_i^* \neq 0 \,\&\, \boldsymbol{P}_{U_i} \neq 1 \\ 0 & \boldsymbol{U}_i^* = 0 \end{cases} \tag{15}$$

With the distribution of observations in the virtual sample set, the distribution of the output of the aggregation function, marked as $p(\mathbf{s}^{B*})$ can be computed with the same methods as Equation 10. As a result, we can skip sampling the virtual sample set and directly draw instances from $p(\mathbf{s}^{B*})$ to get the results of the aggregation function, which is similar to the kernel trick in kernel methods, so we name it Pseudo Resampling Trick.

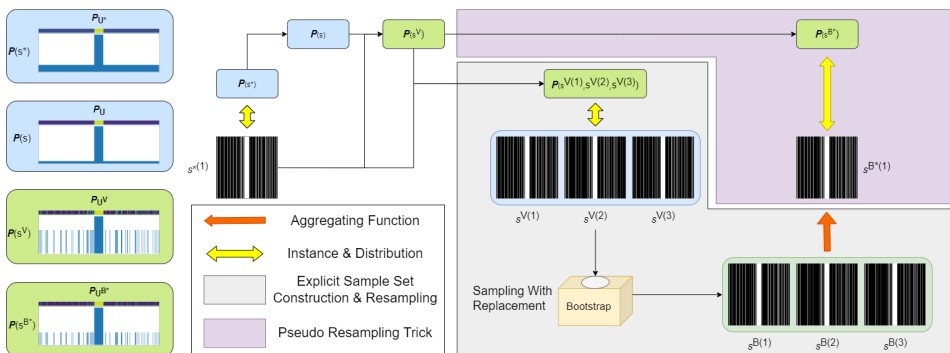

Figure 3: Demonstration of virtual sample set and pseudo resampling trick. The virtual sample set of an observation and its corresponding distribution are visualized. The gray area is the explicit construction of the virtual sample set and conducting Bootstrap sampling, while the purple area corresponds to the pseudo resampling trick.

## 4.3 SMARTER RE-UNDERSAMPLING AND TRAINING WITH BOOTREC

It's easy to find that $\{i | \boldsymbol{U}_i^{B*} = 1\} \subseteq \{i | \boldsymbol{U}_i^* = 1\}$, so the process results in re-undersampling in k-space. On the contrary, for any given re-undersampling mask $\boldsymbol{U}^R$ applied to the sample, the process can be described as the pseudo resampling virtual sample set, as long as the distribution of $\boldsymbol{U}^{R*} = \boldsymbol{U}^R \odot \boldsymbol{U}^*$ is the same as $\boldsymbol{U}^{B*}$. Based on this insight, new algorithms can be developed to

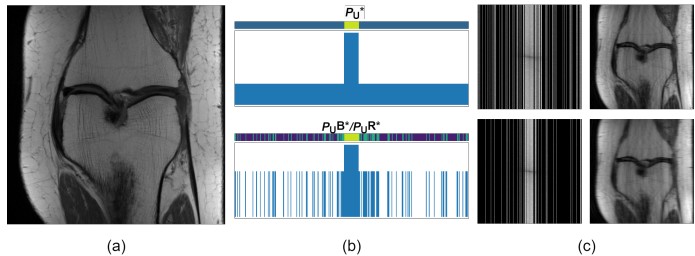

Figure 4: Example of dataset used. (a): A fully sampled image. (b): The upper is the distribution of $P_{U^*}$ and the lower is $P_{U^{B*}}$ or $P_{U^{R*}}$. (c): The (re-)undersampled k-space and corresponding images.

implement the Bootstrap computation like Equation 13. The pseudo-code is provided in Appendix B.

A further step is to use estimated MSE as a proxy for the loss function in training learning-based reconstruction models. Algorithm 2 can derive the loss function as Equation 16 and Appendix D provide an example pipeline of training.

$$\mathcal{L}_{bootrec}(\hat{\boldsymbol{x}}, \boldsymbol{y}^*) = \hat{mse} = \frac{1}{m} \sum_{k=1}^{k=m} (\hat{\boldsymbol{x}}^{B_k} - \hat{\boldsymbol{x}})^2 \tag{16}$$

The key attributes of our methodology include:

1. The re-undersampling pattern is derived from the distribution of sampling and is dynamic[1].
2. Spatial loss is used and the final target of $\hat{\boldsymbol{x}}$ is involved in the training process.
3. The self-supervision loss can be interpreted as errors estimated.

## 5 EXPERIMENTS

### 5.1 IMPLEMENTATION METHODS AND BASELINES

The experiment is conducted in fastMRI dataset (Zbontar et al., 2018), applying the setting of Wang et al. (2022b), where 232 volumes are split into 2D slices and divided with around 8:1:1 for training, validation, and testing, with a sampling ratio of 33% and a fixed mask for all data points. Coil sensitivity maps are built with ESPiRiT (Uecker et al., 2014). We use a large hyper-parameter $n = 1000$, 100 epochs for training models, and 100 iterations in each zero-shot epoch. More details are in Appendix F.

### 5.2 EVALUATING ESTIMATED MSE AS UNCERTAINTY QUANTIFICATION

To test the effectivenAlgorithm 2 in an independent reconstructor, a DC-CNN model (Schlemper et al., 2018a) is trained with supervised MSE loss. The visualization of the results is in Figure 5. The data are collected with the test set so the model doesn't meet them in training or validation. We evaluate the correlation between the estimated MSE and the ground truth computed from the label and the prediction and the influence of different $m$ values. The correlations of the estimated and ground truths mean that it's possible to identify hard sample with the estimation.

### 5.3 ESTIMATED MSE AS LOSS FUNCTION FOR SELF-SUPERVISED TRAINING

We test the capability of optimizing reconstruction model according to Bootstrap estimated MSE in the zero-shot reconstruction scenery (Yaman, 2022), where an untrained neural network is optimized

---

[1]In fact, some intuition of selecting such masks can also be derived, such as the re-undersampling ratio should not be lower than $\lim_{n \to +\infty} 1 - (1 - 1/n)^n = 1 - \frac{1}{e}$, otherwise it would result in a negative cardinality of the virtual sample set. Another situation is the corresponding $n$ is not integer.

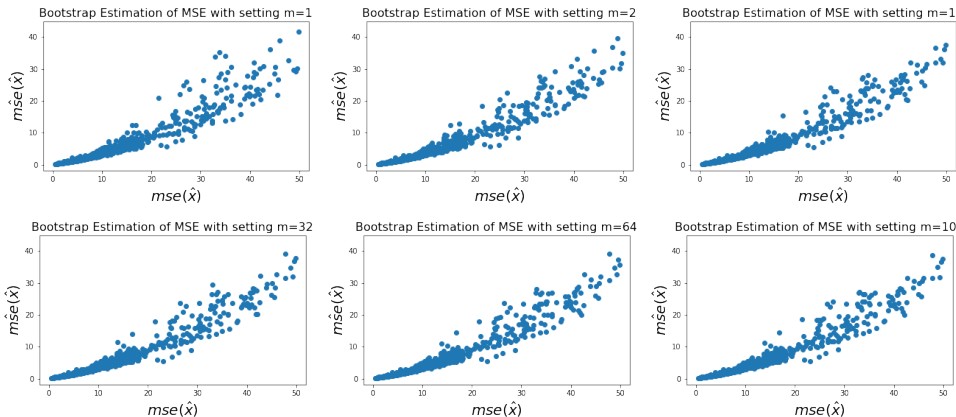

Figure 5: Estimated MSE with Algorithm 2 v.s. MSE computed with the ground truth. All figures show clear linear correlations between the estimations and ground truths, and the different values of $m$ seem to have little influence on the correlation.

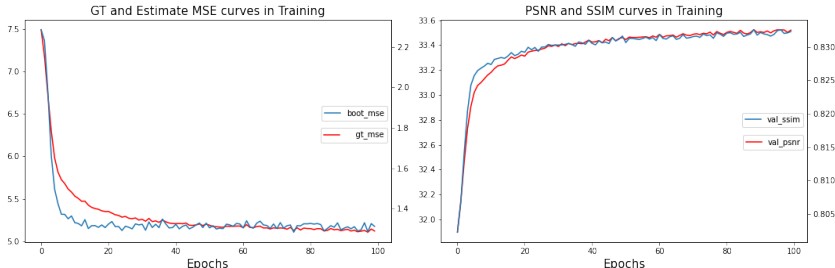

Figure 6: Training curves in zero-shot training. Estimated and GT MSE show consistent tendency, proving the effectiveness of optimizing Estimated MSE.

according to the acquired data with some loss function. Note the optimization of the loss function cannot be conducted directly with gradient descent, since the model will collapse. Accordingly, we make some special designs including stopping the gradient of the original estimation(Grill et al., 2020; Chen & He, 2021) and enforcing consistency on positions not sampled. The details of the implementation can be found in Appendix D. We set $m = 1$ in the experiment.

We use data from the fastMRI validation set. Our method is compared with other zero-shot methods like deep image prior (Ulyanov et al., 2018; Jafari et al., 2021; Senouf et al., 2019) and self-supervision via data undersampling (SSDU) (Yaman, 2022), where independent zero-shot models trained separately per image. Details of the models and results can be found in Appendix E. We also show the performances of SENSE reconstruction (Pruessmann et al., 1999), supervised DC-CNN methods, and the state-of-the-art (SOTA) unsupervised model from Wang et al. (2022b)[2]. The quantitative results can be found in Figure 7 and visual examples are displayed in Figure 8.

**Zero-shot model performances** In Figure 7 we show that pure k-space loss functions failed to perform much better than the simple zero-filled SENSE method in our implementation, while our model shows clear advantages. We also probe the metrics during training in Figure 6 and found that the model continuously performs better as the optimization goes. Another positive finding is the MSE of the reconstruction shows a synchronous tendency as the Bootstrap estimated values, proving the accuracy of the estimation.

**Handling multiple observations** Another intriguing prospect is generalizing the loss to the training over multiple samples in a dataset. For now, our theory doesn't directly cover the multi-sample situation. If multiple samples are trained, the distribution is compositional and the to-be-estimated

---

[2]This model binds another more powerful backbone than DC-CNN in training so may have extra advantages.

parameter is transferred to be the parameter of the distribution of $x^{(i)}$. Since the pseudo resampling and other tools are defined to be applied only for $x$ (here it only means a particular image), part of our theory needs to be re-explained and we leave it for future study. However, we demonstrate the effectiveness of directly applying Algorithm 2 in multiple samples. The model trained with multiple samples has better performances than the zero-shot models and even has a better structural similarity index (SSIM) than the SOTA unsupervised model while having a competitive peak signal-to-noise ratio (PSNR).

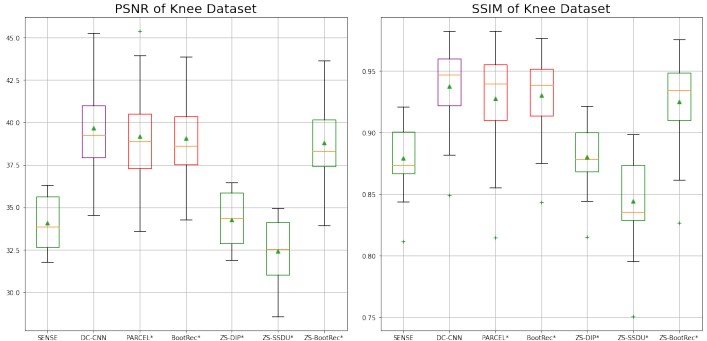

Figure 7: PSNR and SSIM of different models. Purple boxes indicate supervised models; red boxes show unsupervised models; green boxes indicate methods without training data.

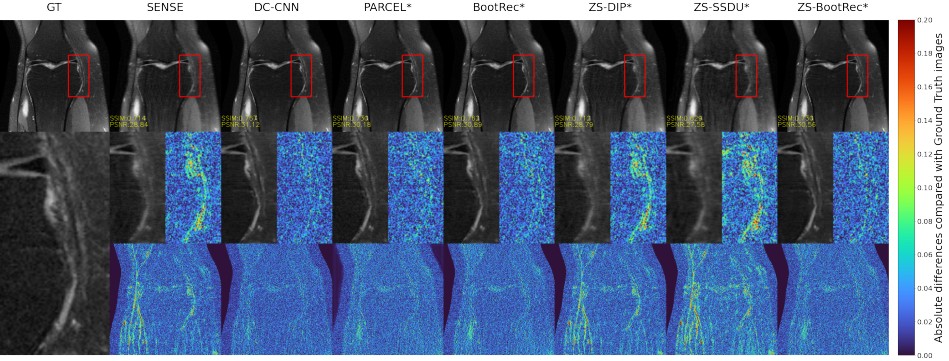

Figure 8: Visual examples of reconstruction of fastMRI validation set. The error maps are amplified by 5 times for better presentation.

## 6 CONCLUSION AND FUTURE WORK

In conclusion, as an attempt to provide a theoretical foundation and direct design of self-supervised learning algorithms, we propose a new paradigm for unsupervised compressed sensing MRI reconstruction. Unsupervised MRI reconstruction is modeled as parameter estimation, then we can wrap existing reconstruction methods to form estimators. Based on this insight, several designs including aggregation function, equivalent sample distribution, virtual sample set, and pseudo resampling trick are proposed to connect re-undersampling in self-supervised learning with Bootstrap sampling. Our flexible framework can not only estimate the MSE of arbitrary reconstructions without accessing ground truth images but also train self-supervised models for better performance.

We believe our paradigm may also inspire some new insights into transforming unsupervised learning. For example, if we define corresponding domains, all augmentations on self-supervised learning may be transformed to re-undersampling and thus can be analyzed with our framework. Also, the proposed aggregation function and estimators are not fully studied and have a large space for improvement with future efforts. The training process of the derived loss function may suffer from collapsing and exploding, which can also be further addressed for better solutions.

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

## A    NOTATIONS

| Symbol | Description |
|---|---|
| $\boldsymbol{x}$ | The fully-sampled spatial MRI image. (Ground truth for supervision. Inaccessible for self-supervision.) |
| $\boldsymbol{y}$ | Undersampled k-space data computed with Equation 2. (The first-hand acquired data in medical practice.) |
| $\boldsymbol{U}$ | Undersampling mask. (Indicating positions to fill in the acquisition of MRI.) |
| $\boldsymbol{s}$ | The observation, composed of $\boldsymbol{y}$ and $\boldsymbol{U}$. (Given information after MRI acquisition.) |
| $\hat{\boldsymbol{x}}$ | The output of the reconstruction model with Equation 3. |
| $.^{*}$ | Values from one-time collected sample set containing only one observation. |
| $.^{(i)}$ | Values in the $i_{th}$ observation of sample set containing multiple observations. |
| $.^{B}$ | Values from the Bootstrap resampled set. $.^{B_k}$ means the $k_{th}$ set. |
| $.^{V}$ | Values from the virtual sample set. |

## B    IMPLEMENTATION OF ALGORITHMS

---

**Algorithm 1:** Pseudo Resampling Function

**Input:** Mask $\boldsymbol{U}^{*}$, Mask probability $\boldsymbol{P}_{\boldsymbol{U}^{*}}$, Size of virtual sample set $n$
**Result:** Bootstrap observation mask $\boldsymbol{U}^{B*}$

1 **def** main$(\boldsymbol{U}^{*}, \boldsymbol{P}_{\boldsymbol{U}^{*}}, n)$ **:**
2     calculate $\boldsymbol{P}_{\boldsymbol{U}}$ according to Equation 11;
3     initialize $\boldsymbol{P}_{\boldsymbol{U}^{B*}}$ as a zeroed tensor with the same shape of $\boldsymbol{P}_{\boldsymbol{U}^{*}}$;
4     fill $\boldsymbol{P}_{\boldsymbol{U}^{B*}}$ with Equation 15;
5     draw random variable $\boldsymbol{V}$ with uniform distribution in $[0, 1)$ and the same shape $\boldsymbol{U}^{*}$;
6     return the 0-1 converted result of $\boldsymbol{U}^{B*} = toInteger(\boldsymbol{V} < \boldsymbol{P}_{\boldsymbol{U}^{B*}})$

---

**Algorithm 2:** Bootstrap Estimation MSE of Reconstruction

**Input:** Observed k-space $\boldsymbol{y}$, Mask $\boldsymbol{U}^{*}$, Mask probability $\boldsymbol{P}_{\boldsymbol{U}^{*}}$, Size of virtual sample set $n$, Number of Bootstrap samples $m$, Reconstructor $f$
**Result:** Estimated MSE $l_{mse}$

1 compute the original estimation $\hat{\boldsymbol{x}} = f(\boldsymbol{y}, \boldsymbol{U}^{*})$;
2 $boot\_mse\_list \leftarrow \emptyset$;
3 **for** $k \leftarrow 1$ **to** $m$ **do**
4     get resampling mask $\boldsymbol{U}^{B_k}$ with Algorithm 1;
5     resample the k-space with $\boldsymbol{y}^{B_k} = \boldsymbol{U}^{B_i}$;
6     compute the corresponding estimation $\hat{\boldsymbol{x}}^{B_k} = f(\boldsymbol{y}^{B_k}, \boldsymbol{U}^{B_k})$;
7     compute the mean square error between the estimation and the original estimation $\hat{mse}^{B_k} = (\hat{\boldsymbol{x}}^{B_k} - \hat{\boldsymbol{x}})^2$;
8     append $\hat{mse}^{B_k}$ to $boot\_mse\_list$ ;
9 average the values to get the Bootstrap estimation of MSE: $mean(boot\_mse\_list)$;

---

## C  DERIVATION OF DISTRIBUTION OF VIRTUAL SAMPLE SET OBSERVATIONS

From an intuitive aspect, for a given position $i$, the size of the virtual sample set $n$ and the corresponding equivalent sample distribution parameter $\boldsymbol{P}_{\boldsymbol{U}_i}$, the times of the position being selected in $n$ observations in the virtual sample set, represented as $\boldsymbol{C}_i$, follows a Binomial distribution $\mathcal{B}(n, \boldsymbol{P}_{\boldsymbol{U}_i})$. Thus, with Bayes' Theorem, the conditional probability can be computed in Equation 17.

$$
\begin{aligned}
&Pr(\mathbf{U}_i^V = 1 | \boldsymbol{U}_i^* = 1) \\
&= \frac{Pr(\mathbf{U}_i^* = 1 | \boldsymbol{U}_i^V = 1)Pr(\mathbf{U}_i^V = 1)}{Pr(\mathbf{U}_i^* = 1)} \\
&= \frac{Pr(\mathbf{U}_i^* = 1 | \boldsymbol{U}_i^V = 1)Pr(\mathbf{U}_i^V = 1)}{1 - (1 - P_{\boldsymbol{U}_i})^n} \\
&= \frac{Pr(\mathbf{U}_i^V = 1)}{1 - (1 - P_{\boldsymbol{U}_i})^n} \\
&= \frac{Pr(\mathbf{U}_i^V = 1 | \boldsymbol{C}_i > 0)Pr(\mathbf{C}_i > 0) + Pr(\mathbf{U}_i^V = 1 | \boldsymbol{C}_i = 0)Pr(\mathbf{C}_i = 0)}{1 - (1 - P_{\boldsymbol{U}_i})^n} \\
&= \frac{Pr(\mathbf{U}_i^V = 1 | \boldsymbol{C}_i = k)Pr(\mathbf{C}_i > 0)}{1 - (1 - P_{\boldsymbol{U}_i})^n} \\
&= \frac{\sum_{k=1}^{+\infty} Pr(\mathbf{U}_i^V = 1 | \boldsymbol{C}_i = k)Pr(\mathbf{C}_i = k)}{1 - (1 - P_{\boldsymbol{U}_i})^n} \\
&= \frac{\sum_{k=1}^{n} \frac{k}{n} Pr(\mathbf{C}_i = k)}{1 - (1 - P_{\boldsymbol{U}_i})^n} \\
&= \frac{\sum_{k=1}^{n} \frac{k}{n} \binom{k}{n} P_{\boldsymbol{U}_i}^k P_{\boldsymbol{U}_i}^{n-k}}{1 - (1 - P_{\boldsymbol{U}_i})^n}
\end{aligned}
\tag{17}
$$

The computation is too complicated and hard to use, so we provide a simplified version by analyzing three special cases:

1. $\boldsymbol{C}_i = 0$, which means that the position is not selected in all observations, thus $\boldsymbol{U}^V$ will always be zero.

2. $\boldsymbol{P}_{\boldsymbol{U}_i} = 1$, common in the MRI acquisition to keep more low-frequency information. In this case, every observation in the virtual sample set contains the position.

3. $\boldsymbol{P}_{\boldsymbol{U}_i}$ is very small. In this case, the probability of being selected again ($\boldsymbol{C}_i > 1$) in the observations is small and can be neglected. If $\boldsymbol{C}_i \neq 0$, we can assume $\mathbf{U}^V \sim \mathcal{B}(\boldsymbol{U}^V; 1, 1/n)$.

Note that only $\boldsymbol{P}_{\boldsymbol{U}^*}$ is given and $\boldsymbol{P}_{\boldsymbol{U}_i}$ is computed with Equation 11, and $n$ is a hyper-parameter that can be set manually, so for better approximation we can choose larger $n$ when $\boldsymbol{P}_{\boldsymbol{U}_i^*}$ is not small enough. As a result, the third case can actually cover all halfway situations by setting proper $n$. Some examples of $C_i$ distributions are shown in Figure 9. It can be found that the conditional distributions (golden bars) have a much higher probability in $\boldsymbol{C}_i = 1$, which means nearly all sampled positions are sampled only once.

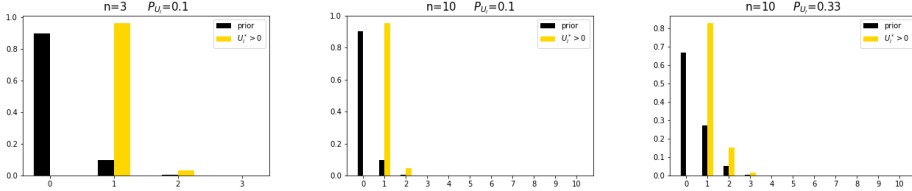

Figure 9: Examples of $\boldsymbol{C}_i$'s distribution with different $n$ and $\boldsymbol{P}_{\boldsymbol{U}_i^*}$. The golden bars show distributions when $\boldsymbol{C}_i > 0$ is conditioned. The corresponding $\boldsymbol{P}_{\boldsymbol{U}_i}$ is around 0.0345, 0.0105 and 0.0393.

Combining the situations, we can get the distribution of the mask.

$$P_{\boldsymbol{U}_i^V} = Pr(\mathbf{U}_i^V = 1) = \begin{cases} 1 & \boldsymbol{P}_{U_i} = 1 \\ 1/n & \boldsymbol{U}_i^* \neq 0 \& \boldsymbol{P}_{U_i} \neq 1 \\ 0 & \boldsymbol{U}_i^* = 0 \end{cases} \tag{18}$$

## D  THE PIPELINE OF BOOTSTRAP SELF-SUPERVISED TRAINING

By referring to Algorithm 2, the Bootstrap estimated MSE can serve as a loss function for self-supervised training. The PyTorch-style code snippet of the training process is shown below. We use stop-gradient and inverse consistency to enable and stabilize the training.

```
n = 5
m = 20

with torch.no_grad():
    x_orginial=model(y*mask,mask)
    y_original=fft(y_original)
for _ in range(m):
    re_mask=generate_re_mask(mask,p_maskin,n)
    x_bootstrap=model(y*re_mask,re_mask)
    y_bootstrap=fft(x)
    x_bootstrap=ifft(y_bootstrap*(~mask)+y_orginial*mask)
    loss=nn.MSELoss(x_bootstrap,x_orginial)/m
    loss.backward()
optimizer.step()
```

## E  BRIEFS OF OTHER ZERO-SHOT MODELS

Besides the basic SENSE reconstruction that zero-fills the missing k-space, the differences between the models mainly come from the different loss functions. All the models are based on the same backbone structure of 8-layer DC-CNN. Note that due to the memory limit, the weight of the model is shared among the z-axis.

For the DIP models, the loss function is limited to keeping the consistency of all the knowing positions. Due to the data consistency layers in the model, the loss in k-space will always be 0 and nothing can be learned, so the loss is defined in the image domain as Equation 19, where $f_{SENSE}$ is the SENSE reconstruction.

$$\mathcal{L}_{DIP}(\hat{\boldsymbol{x}}, \boldsymbol{y}^*) = mse(f_{SENSE}(\boldsymbol{y}^*), \hat{\boldsymbol{x}}) \tag{19}$$

The SSDU model applies re-undersampling in training and defines a normalized $l_1 - l_2$ loss in k-space as Equation 20. As for the super-parameters, we choose $K = 10$ and the re-undersampling ratio of $40\%$ following the optimal setting of original paper(Yaman et al., 2020), and a uniform re-undersampling strategy is used. Also, Gaussian re-undersampling is conducted as described in the paper instead of deriving re-undersampling masks based on virtual sample set. The settings are modified to follow our basic setting of sharing weights among different slices.

$$\mathcal{L}_{SSDU}(\hat{\boldsymbol{x}}, \boldsymbol{y}^*) = \frac{||\boldsymbol{y}^* - \mathcal{F}\hat{\boldsymbol{x}}||_2}{||\boldsymbol{y}^*||_2} + \frac{||\boldsymbol{y}^* - \mathcal{F}\hat{\boldsymbol{x}}||_1}{||\boldsymbol{y}^*||_1} \tag{20}$$

## F  IMPLEMENTATION METHODS

Given a reconstructor $f$ and an acquisition $\boldsymbol{y}^*$ and $\boldsymbol{U}^*$ of the spatial anatomy $\boldsymbol{x}$, a reconstruction and estimation result can be obtained by $\hat{\boldsymbol{x}} = f(\boldsymbol{y}^*, \boldsymbol{U}^*) = f_{AF}((\boldsymbol{y}^*, \boldsymbol{U}^*))$. MSE and its provide a simple but effective evaluation of the differences between $\hat{\boldsymbol{x}}$ and the target $\boldsymbol{x}$.

To conduct the bootstrapping in reconstructing a single MRI observation, although it may be complicated to state the transformation of the problem, the resulting solution is rather simple and intuitive. The construction of a virtual sample set and the computation of aggregation function become invisible parts of the pipeline. The only extra thing we need to do is to compute a re-undersampling mask based on the selected cardinality $n$, the actual mask $U$ and its distribution parameter $P_U$. We provide the pseudo code of the computation in Algorithm 1. As bootstrapping enables the computation of various statistic properties without acquiring more sample sets, statistics like MSE can be estimated with only a single observation of partial measurements acquired CS MRI. The pseudo code of unsupervised computation of MSE can be found in Algorithm 2 based on Equation 13.

It can be seen that besides the input and the reconstructor, the most important hyper-parameters of the algorithms are the cardinal of the virtual sample set $n$, which decides the ratio and positions of re-undersampling and the times of Bootstrap sampling $m$, which increases the accuracy of Bootstrap estimators as it grows.

## G    QUANTITATIVE COMPARISON OF UNCERTAINTY QUANTIFICATION

With residual errors as UQ notion, we compare our methods with residual magnitude prediction(Angelopoulos et al., 2022), a method that directly enables a separated prediction head to predict the errors and we name it Residual Regression (RR) for simplification. The model is trained on fastMRI single-coil dataset with the same setting from the original paper except for a smaller batch size (48 v.s. 78) due to the limited GPU memory. After training, we directly apply Algorithm 2 of BootRec in the trained model to get our quantification, and square the predictions from RR. We also apply Stein's Unbiased Risk Estimator (SURE) to estimate the trained model's MSE, with settings from (Edupuganti et al., 2020).

Evaluation of UQ is still an open problem, with various metrics focusing on different notions and aspects. Considering the task, we conduct inter-image evaluations and intra-image evaluations. For inter-image evaluation, we average the pixels' estimated MSE for every image and collect the ground truth MSE (GT MSE) of the reconstruction. The visualization can be found in Figure 10 and the quantitative results of correlation coefficients are summarized in Table 1. It can be seen that BootRec estimated MSE continuously presents better coefficients compared with predictions from RR head and has the highest values in Spearman and Kendall Coefficients. For intra-image evaluation, the distribution of the errors is more important than the absolute values as it indicates areas of hallucination. For the inter-image evaluation, we display an example from validation set for comparison of error distributions in Figure 11. The GT MSE shows high error levels in border areas, which is reasonable as the pixel values change rapidly, and the Bootstrap estimated error shows a similar distribution. The RR and SURE error maps put too much uncertainty inside the tissue, especially for the RR results. SURE estimations performed well in inter-image correlation but failed in inter-image distribution of errors, with one possible explanation that the residual variance of specific position is hard to compute.

|  | Pearson | Spearman | Kendall |
|---|---|---|---|
| RR | 0.866 | 0.864 | 0.692 |
| SURE | **0.941** | 0.752 | 0.561 |
| Bootstrap | 0.888 | **0.920** | **0.756** |

Table 1: Correlation Coefficients computed corresponding to Figure 10. All values have a negligible p-value (we get zero in computation). The Bootstrap estimator achieves best correlation in Spearman and Kendall Coefficients.

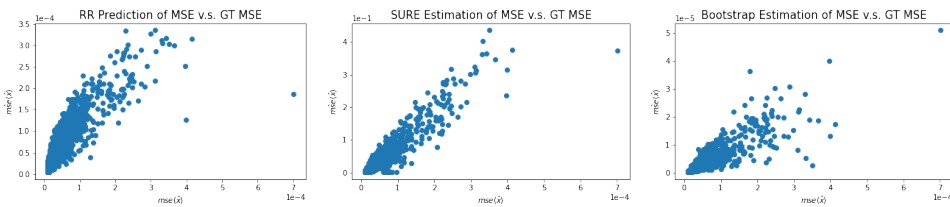

Figure 10: Inter-image Correlation of predicted/estimated MSE in the validation set.

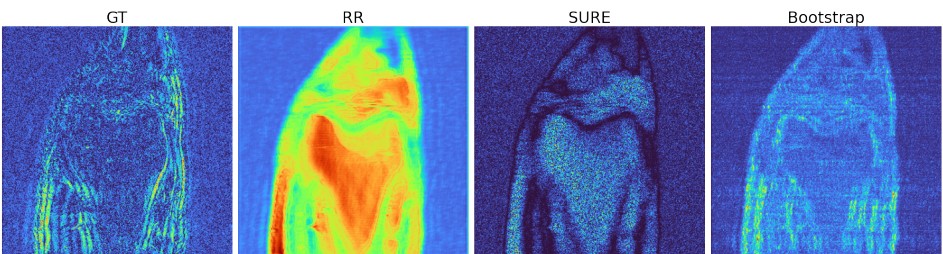

Figure 11: Intra-image distribution of predicted/estimated errors in the validation set (square roots are computed for MSE results). Note that the values are normalized across the image.

