# OpenReview forum: "Self-supervision Meets Bootstrap Estimation: New Paradigm for Unsupervised Reconstruction with Uncertainty Quantification"
_ICLR.cc/2024/Conference — Submitted to ICLR 2024_

### Official Review · Reviewer_FBWg · 2023-10-26

**Soundness:** 3 good
**Presentation:** 3 good
**Contribution:** 3 good
**Rating:** 5
**Confidence:** 2

**Summary:**

This paper proposes a novel self-supervised MRI reconstruction paradigm in which the reconstruction process is modeled as parameter estimation, leveraging the idea of bootstrapping. To mitigate the high variance incurred by randomly generating a sample set, this paper proposes to get the distribution of the observations by mapping the observation to a virtual sample set. The authors conduct both theoretical and empirical analyses for the proposed method.

**Strengths:**

1. The idea is novel and interesting.
2. The demonstrated equivalence between the sample set bootstrapping and the re-undersampling is inspiring.
3. The results are promising to an extent.
4. The paper is well written.

**Weaknesses:**

1. It is highly expected that the paper includes a comparison with other typical methods for MRI reconstruction, such as [1] and [2]. Although the authors argue that deep learning models may suffer from unreliability, as the fast growing of vision transformers, their appealing performance should not be ignored. Thus, a comparison with such methods will make the results more convincing. Although these methods may not focus on zero-shot, the testing performance can still be compared.

2. I think the paper lacks a discussion of the differentiability of the aggregation function h. Although the classical MSE loss is differentiable, due to the use of the aggregation function in the MSE which is adopted to train a model, the authors are highly encouraged to discuss the differentiability of the aggregation function.

Reference

[1] https://proceedings.mlr.press/v172/lin22a/lin22a.pdf
[2] http://proceedings.mlr.press/v139/fabian21a/fabian21a.pdf

**Questions:**

1. Is the uncertainty in this paper partially from the resampling operation due to the use of bootstrap? More insights will be helpful.

2. Although reconstruction is an important topic in MRI, are there any other reasons that make the proposed method tie to MRI reconstruction? In other words, whether the proposed method is suitable for potential reconstruction tasks in natural image domains?

3. Is that possible to use the Monte Carlo method in this work?

---

> ### Author Response · Authors · 2023-11-19
>
> We are really grateful for your recognition of our work and contributions. Below, we address your suggestions and comments.
>
> **Reply to Weaknesses**
> - While we would love to add more comparison with other advanced MRI reconstruction methods, unfortunately, it requires substantial time and access to their original codes (some are not shared though) for an informed and detailed comparison. We add citations to these referred papers, and will further explore them in our future work.
> - The differentiability of the aggregation function becomes a main concern in our derivation but we found it operable for  backpropagation and thus it won’t influence the optimization. For example:
>    - With or without the aggregation function, the gradients of the reconstruction model parameterized by $\theta$ are always:
> $$\frac{\partial{\mathcal{L}}}{\partial{\theta}}=\frac{\partial{\mathcal{L}}}{\partial{f_{AF}}}\frac{\partial{f_{AF}}}{\partial{f}}\frac{\partial{f}}{\partial{\theta}}=\frac{\partial{\mathcal{L}}}{\partial{f}}\frac{\partial{f}}{\partial{\theta}}$$
>    - As for the gradients of $y$ and $U$, we add an explanation after Equation 7, that aggregation function results in a re-undersampling mask which may not influence the differentiability to $y$. The gradients of $U$, on the other hand, may be influenced, but they are not optimized for MRI reconstruction, thus it may not a problem.
>
> **Reply to Questions**
> - In the original version of our paper, uncertainty has been limited to the randomness of undersampling, while the Bootstrap resampling actually “simulates” it. It’s true that for the reconstruction of re-undersampled images, Bootstrapping incorporates uncertainty, but in inference, no re-undersampling will be conducted.
> - We’re extending our work as we discussed in Conclusion. There would be a trivial extension that applies virtual sample and pseudo resampling in similar tasks related to Compressed Sensing. Other image-to-image translation tasks like super-resolution and deblurring could also benefit from our work by defining virtual sample sets in particular domains. We’re open to further comments.
> - At this stage, Mote Carlo method would not be appliable as it incorporates sequential sampling and requires extra control for the  re-undersampling purpose, etc. We will further explore this idea.
>
>
> Thank you very much and please let us know if you have any further comments.

---

> > ### Comment · Reviewer_FBWg · 2023-11-20
> >
> > The authors’ response addresses most of my concerns. However, I still think it would be more convinced to include comparisons with other advanced MRI reconstruction methods. I know adding this would be extremely challenging due to the short rebuttal window, but I think it is very necessary to include them in the future revisions. Seems the authors also plan to do so.

---

### Official Review · Reviewer_n9iq · 2023-10-27

**Soundness:** 2 fair
**Presentation:** 3 good
**Contribution:** 2 fair
**Rating:** 3
**Confidence:** 5

**Summary:**

This paper interprets a popular self-supervised MRI reconstruction approach: self-training with secondary undersampling [1,2], as bootstrapping. The repeated secondary undersamplings are modeled as a virtual sample set, and the secondary sampling masks are aggregated as a sampling distribution for bootstrapping. The authors then propose to use the bootstrapped error to estimate the true underlying reconstruction error. Qualitative results are shown on public MRI datasets and some correlations between estimated errors and the true underlying errors can be observed.

**Strengths:**

The problem of estimating the error in MRI reconstruction is of practical value. It links to the trustworthiness of deep learning based image reconstruction.

The authors link the secondary undersampling technique [1,2] to bootstrapping. This is a very interesting interpretation.

Correlations between estimated reconstruction errors and the true underlying reconstruction errors can be observed.

**Weaknesses:**

An important prior work [3] for modeling aleatoric and epidemic uncertainties for deep learning MRI reconstruction is missing.

The detailed approach and its mathematical framework of the prior works: self-supervised MRI reconstruction using self-training on secondary undersampling [1,2], need to be introduced in Sec. 2, as they are the basis for the entire manuscript.

The manuscript in general lacks clarity: it is difficult to find the key arguments and the core take-home information from the abstract and the introduction.

The writing style is also sloppy with key concepts arbitrarily named, used, but left unexplained. E.g., the first paragraph of Sec. 3.2: the narration is quite casual. Also, what does the paragraph under Eq. 8 mean? What does the starting paragraph in Sec. 4.3 mean? This sloppy writing may make readers who are not familiar with secondary undersampling based MRI reconstruction, extremely difficult to follow. The writing does not meet the high standards of ICLR.

Despite the interesting interpretation of bootstrapping, the manuscript does not make significant theoretical/methodological breakthroughs beyond the existing secondary undersampling based MRI reconstruction approaches [1,2], not to mention that secondary undersampling is not the only approach for unsupervised/zero-shot MRI reconstruction and/or error estimation.

Sec. 5.2: There is a lack of quantitative evaluation of the quality of error estimations. The authors also fail to compare with the well-established Bayesian deep learning based image reconstruction [3]. Notebly, unlike the proposed approach, Bayesian deep learning allows to explicitly separate aleatoric uncertainty and epistemic uncertainty.

[1] https://arxiv.org/abs/2102.07737

[2] https://www.sciencedirect.com/science/article/abs/pii/S1361841522001852

[3] https://link.springer.com/chapter/10.1007/978-3-030-00129-2_8

**Questions:**

Is the most fundamental assumption mentioned at the starting of the manuscript: modeling U as independent Bernoulli’s, unrealistic? In practice sampling patterns are subjects to the physical constraints of the gradient system of the scanners, and the resultant sampling patterns (both original and secondary) are by no means independent.

The authors are suggested to improve the clarity of writings and illustrations.

**Details Of Ethics Concerns:**

This study is mostly based on public datasets and the authors need to check the terms and conditions by the owners of the datasets.

---

> ### Author Response · Authors · 2023-11-19
>
> Thanks for your detailed comments. In response, we provide explanations below.
>
> **Response to Weakness**
> - Thank you for pointing out the missing reference. In this revision, we add the discussions about it.
> - Figure 1 presents the pipeline, and we add math equations to Section 2. We further add brief introduction to some SSR methods to Appendix E.
> - The paper is substantially reorganized responding to the related comments, including enhancing clarity and explanations. For take-home information, we argue and evidence that self-supervised MRI reconstruction can be modeled as error-oriented parameter estimation, accordingly we introduce a novel method - Bootstrap estimation for SSR (BootRec).
> - We polish the paper per your related comments.
>
> About the contribution of our work:
> -  We emphasize the key attributes of our methodology in a new section - Section 4.3.
> -  As mentioned in the introduction and related work parts, the existing methods are mostly heuristic in their design. Our method makes a meaningful trial to answer the following questions with reasons beyond "it just works":
> - Why use **this re-undersampling mask** instead of others?
> - Why use **this loss function** instead of others?
> - The re-undersampling methods, though not the only methods in SSR, it outperforms other models in similar settings [1,2], and our methods are also compared with other types of unsupervised approaches like Deep Imaging Prior.
>
> - Thank you for providing useful information on MRI UQ. The uncertainty quantification part in our work is not the focused claim, we instead focus on the estimation of errors, so the notions become different from [3]. We instead compare our methods with error-oriented UQ in Appendix G, which shares the idea of training a separated network or module to predict uncertainty. Besides, although our methods do not include epistemic uncertainty in the design, it can cooperate with any such methods, including the Bayesian Neural Network in [3], and it is training-free. A detailed comparison and assessment in terms of UQ goes beyond this paper, thus we focus on reconstruction.
>
> **Response to Questions**
> - It’s a very good question to challenge the distribution of U. While the assumption of Bernoulli is a common deed in MRI communities, like the diagonal covariance matrix assumption in UQ and other tasks, it’s not satisfactory. Also, a 1D Cartesian mask may be ideal in the dimension of sampling, and in other cases, different points do have correlations. However, even if the sampling points are not independent, we may always find some “nearly independent elements” in the acquisition, such as the radial tracings in radial sampling or to conduct methods like whitening to reduce the correlation. As for the re-undersampling pattern, we believe we can guarantee independence.
> - The paper is reorganized according to the related comments.
>
> [1] Zero-shot self-supervised learning for mri reconstruction. In International Conference on Learning Representations, 2022.
>
> [2] Parcel: Physics-based unsupervised contrastive representation learning for multi-coil mr imaging. IEEE/ACM Transactions on Computational Biology and Bioinformatics, 14(8):1–12, oct 2022b. ISSN 1557-9964. doi: 10.1109/TCBB.2022.3213669.
>
> [3] Bayesian Deep Learning for Accelerated MR Image Reconstruction. MLMIR 2018
>
> [4] Uncertainty Quantification in Deep MRI Reconstruction

---

> > ### Comment · Reviewer_n9iq · 2023-11-21
> >
> > I would like to thank the authors for the substantial revision of the paper. Some of my concerns are addressed but others remain:
> >
> > - Major contributions: resampling-based SSL reconstruction itself (by Yaman et al.) might not be counted as a major contribution of your work. Instead, if I interpreted correctly, the major contribution should be its bootstrapping interpretation, in together with its utility for error quantification.
> >
> > - I am not fully convinced about the reason why estimating MSE would bring more clinical benefit than estimating aleatoric and epistemic uncertainties? Given that these two uncertainties have straightforward real-world interpretations of different aspects/sources of uncertainty.
> >
> > - I am still not fully convinced why error quantification for a very specific algorithm (k-space re-sampling) for a very specific task (self-supervised learning for deep learning MRI reconstruction) would be of sufficient interest to most of the readers of ICLR, despite that re-sampling may be the state-of-the-art approach for that very specific task for now. What do the authors think to be the broader impact of the proposed work?
> >
> > - Appendix G should be put forward into the main text.

---

> ### Author Response · Authors · 2023-11-22
>
> Thank you for your kind response.
>
> - The interpretation is among the key aspects of our work. Indeed, we never claim the contribution of proposing re-undersampling or introducing self-supervision in MRI reconstruction.
> - The field of uncertainty quantification is not very familiar to us so I may not be qualified to judge different uncertainty notions in this paper. In our observation, the errors seem to be more explicit and easier to judge since we can get the GT values. Further studies will be beneficial and thanks to your valuable perspective.
> - Please be aware that our quantification leverages re-undersampling but is not limited to SSR tasks (Section 5.2 and Appendix G conduct estimations on arbitrary models). In fact, we observe minimal constraints associated with applying our methodology and find broader applications (see our discussion with reviewer FBWg) in self-supervised learning of image-regression tasks, though with limited space to extend the discussion.
> - We'd make a final revision of our paper if possible.

---

### Official Review · Reviewer_4bq6 · 2023-10-30

**Soundness:** 2 fair
**Presentation:** 2 fair
**Contribution:** 2 fair
**Rating:** 3
**Confidence:** 4

**Summary:**

The paper titled: Self-supervision Meets BOOTSTRAP Estimation: New Paradigm for Unsupervised Reconstruction with Uncertainty quantification proposes a novel SSR methods with BootStraping for MRI reconstruction, with the ability of uncertainty estimation and quantification.

**Strengths:**

1. Uncertainty quantification for MRI reconstruction is an open problem, based on my knowledge, I think this paper is the first one to quantify MRI uncertainty in an unsupervised or self-supervised manner from under-sampled MRI.

2. I like the idea of using BootStrap to resample the undersampling pattern, and from the Figure 7 plots, the results outperform SSDU and other SOTAs in terms of quantitative metrics.

3. The algorithm is well-written and delivered.

**Weaknesses:**

1. I think one of my biggest concerns is how the proposed method compared with other existing approaches for uncertainty quantification, there have been a wide range of works on this topic, they are either sampling based [Uncertainty Quantification in Deep MRI Reconstruction] or directly estimation the absolute residual error [Rigorous Uncertainty Estimation for MRI Reconstruction], this paper lack the comparisons with other approaches, please discuss/cite them.

2. For the reconstruction results, the authors only showed an PSNR and SSIM plot (Figure 7) without any visual results to inspect on the details, the only visual results is Figure 5, which also doesn't deliver much information. I think this paper demonstrates a proof-of-concept, but lack evaluations.

3. What is the purpose of estimating MSE, this can be generalized to an open question, how to use the uncertainty estimation results for diagnosis, I can imagine it would be useful if we compute uncertainty in latent space, but could you elaborate on how to use uncertainty estimation for down-stream task?

One inspiring paper of quantifying theoretical results of uncertainty estimation is: [https://arxiv.org/abs/2202.05265 Image-to-Image Regression with Distribution-Free Uncertainty Quantification and Applications in Imaging], you maybe able to get some insights from.

**Questions:**

1. How to quantitatively evaluate the quality of your uncertainty estimation results.

---

> ### Author Response · Authors · 2023-11-19
>
> Thank you for your recognition of the novelty of our work, as well as to your helpful suggestions.
>
> As we clarify in the summary of changes to all reviewers, our focus of this work is not on the Uncertainty Quantification (UQ). In the revision, we reorganize the paper for more focus on proposing a novel reconstruction method. Below, we address your specific comments.
>
> **The evaluation of UQ**
> - We add more discussion on UQ in the section on Related Work while focusing our quantification on MSE estimation in UQ. We also discuss the work you referred to. The UQ in our work is specifically referred to error estimation.
> - We compare the estimation of MSE with residual magnitude regression of the recommended work [Image-to-Image Regression with Distribution-Free Uncertainty Quantification and Applications in Imaging] (which should cover the family of similar methods) and SURE estimation of [Uncertainty Quantification in Deep MRI Reconstruction] in Appendix F. Unfortunately we cannot compare our work with the method in [Rigorous Uncertainty Estimation for MRI Reconstruction] because we do not have access to that paper's materials. We are investigating the implementation of more methods and further analysis may also be added later.
> - For evaluation metrics, we find the correlation between the estimated and actual MSE serving as a good indicator for MSE-based UQ, since the ground truth is clear. Other existing evaluation measures mainly focus on variance and quantiles, so we are not able to use them.
>
> **Visual Presentation of Reconstruction Results**
> - Visual examples including the reconstruction results and error maps are added to Figure 8 in the revised paper.
>
> **The application of UQ**
> - Specifically for our work, the purpose of estimating MSE has a rather simple and special answer: that is to optimize it. That’s the core idea of our proposed self-supervised MRI reconstruction method, please refer to Section 5.3. Please be noted, this may not be generalized to other UQ notions or methods since they may either need training itself or be not suitable for optimization.
> - The application of general UQ is worthy of discussion. We add a brief discussion at the beginning of Section 2.3a.

---

### Official Review · Reviewer_tf3c · 2023-10-31

**Soundness:** 2 fair
**Presentation:** 2 fair
**Contribution:** 2 fair
**Rating:** 3
**Confidence:** 5

**Summary:**

This paper introduces a novel self-supervised approach for MRI reconstruction using bootstrap sampling. The method involves re-subsampling the undersampled k-space, reconstructing each resampled measurement, and formulating a loss function based on the reconstruction of the original measurement and the mean squared error of the resampled reconstruction.

**Strengths:**

The approach presents an innovative concept and demonstrates commendable performance. Notably, the training loss trajectory aligns consistently with that of training on self-supervised MSE, as shown in Figure 6.

**Weaknesses:**

The primary challenge with this paper lies in its presentation, making it difficult for readers to follow. Some issues include:
- Excessive Equations and Notations: The paper is overwhelmed with equations and notations, overshadowing the fundamental concept. The core idea appears to be sampling, but the multitude of equations adds unnecessary complexity without aiding comprehension.
- Confusing Notations: Notations like the two 'U's in Equation (6) are ambiguous and visually similar, leading to confusion and hindering understanding.
- Unexplained Figures: Figures, such as Figure 4, lack detailed explanations, leaving readers without essential context to interpret the visual data.
- Lack of Explaining Prior Works: Assumptions about the reader's familiarity with existing work, especially (Yaman, 2022)'s zero-slot learning, create gaps in understanding. The absence of pertinent details hampers comprehension.

I suggest the authors clean up the notation, reformat this paper, add more details, and make resubmission to another conference.

Another concerns the authors might take into consider for improving this paper:
- Unclear Significance of Variance: The paper lacks clarification on why the variance (uncertainty) of bootstrap resampled reconstruction is important. Address the relevance of this aspect, especially in comparison to uncertainty quantification for raw measurement reconstruction.
- Theoretical Foundation: While the paper claims to provide a "theoretical foundation," it predominantly relies on equations without substantive theoretical analysis. A more comprehensive exploration of the theoretical underpinnings is essential to substantiate this claim.

**Questions:**

See the weakness part.

---

> ### Author Response · Authors · 2023-11-19
>
> We appreciate your comments. We made major revisions to address your comments and enhance the paper's readability. Here we provide the point-to-point responses.
>
> **Reply to Weaknesses**
> - We add a note to the Introduction section directing readers to the table of notations in the appendix. The equations in the main content are largely reduced as well (from 25 to 16).
> - Do you mean the Us in Equation(5)? The two $$U$s refer to the “random variable” and “vector value” of the mask $U$ with display styles provided by the ICLR formatting instructions. Anyway, we remove the equations.
> - We add explanations to the figures. Specifically, for Figure 4 in the original submission (i.e., Figure 5 in this revision), it shows the correspondence of the estimated results v.s. the actual values.
> - We add a new figure - Figure 1 - to summarize the framework of MRI reconstruction, which may help explain our work better. The detailed settings of the previous methods are provided in Appendix F.
>
> **Further replies to the concerns:**
> - We agree and notice the inclusion of variance estimation becomes redundant and compromises the integrity of the article. Such discussion may be pertinent to another independent research.
> - We will work on more theoretical analysis.
> - The abstract is revised.
>
> We would greatly appreciate if you could kindly revisit the entire manuscript at your convenience, in particular, check our significant revision. Please let us know if you have any further comments.

---

> > ### Comment · Reviewer_tf3c · 2023-11-22
> > **Thanks for the rebuttal**
> >
> > I would like to thanks for the author's efforts to improve the manuscript.
> > - **Presentation**: I think, after the revision, the presentation of this paper has been improved. Yet, I believe the authors could do a even better job if having more time than this discussion period. More suggestions: (a) In the new Figure 1, it is better to also include your own method; (b) It is easier for me to understand this paper by looking at the Appendix B and D, rather than the main paper. I think there is still room for the improvement of the presentation (in the main paper).
> > - **Variance estimation**: Thanks for your response.
> > - **Theoretical foundation**: When I wrote this comments, I did not mean that this paper must having a theoretical analysis. Many good papers do not have theoretical analysis. I just wanted to highlight that, if you don't provide valuable theoretical results, don't over-claim it in the paper.
> >
> > I will raise my score to reflect the authors' effort in this rebuttal. Thank you!

---

### Author Response · Authors · 2023-11-19
**Summary of the significant changes made in the revision**

We thank all reviewers for your comments and suggestions. By reading the comments, on one hand, we appreciate your valuable suggestions, which are addressed in the revision. On the other hand, we also realize some potential misreading or misunderstanding. To improve the paper's readership, in the revision, we've made major modifications which are summarized below. In addition, we provide specific responses to the corresponding reviewers.

We find many comments and concerns leaned to the uncertainty quantification part of our work. In fact, our work is primarily intended to transform self-supervised MRI reconstruction algorithms. Therefore, connecting variance estimation to uncertainty quantification is a secondary result of our work, that’s why we use “with” instead of “and” in the title).

By referring to extra literature, including those recommended by Reviewers #4bq6 and #n9iq, we understand the reviewer's perspective on the methodology and concepts of Uncertainty Quantification, and thus make further clarification accordingly.

As our focus is on self-supervised MRI reconstruction, without losing the integrity of our work in addressing the above concern, we  make major revisions in the revised paper, which include:
- We eliminate the content about the estimation of variance since they are not included in the self-supervised training.
- The estimation of MSE is connected to error-based uncertainty notions, and UQ mainly serves as an intermediate stage before we reach self-supervised learning.
- We add more discussion on the Uncertainty Quantification and its related methods.
- Quantitative comparison with the existing UQ method is conducted.

Further, we reorganize the framework of our method to make it easier to follow. Necessary background introduction and clarification are also added.

As this revision involves significant changes, we would appreciate if the reviewers could take another look at the revised paper and let us know if you have any further concerns or comments.

---

### Author Response · Authors · 2023-11-19
**Extended Experiments on Variance Estimation**

To address reviewer #tf3c's comment on the significance of variance, due to space limitation, while we do not include the discussion on variance estimation in the paper, we have some quantitative experiment results on the variance-based UQ. Below, we provide these results for your reference.

All the compared methods are from reference [1], the results are direct output of the uncertainty parameters from the neural network. The models are trained with their official settings on fastMRI single-coil reconstruction, except for smaller batch sizes. We directly apply Bootstrap Variance Estimation on the trained network to estimate the variance of the trained model with $m=5$. The evaluation is based on the code from the Uncertainty Toolbox [2]. For the Quantile Regression model and Residual Regression model, the check score and interval score are computed with interpolation of quantiles where the latter is considered to form an interval with the predicted residuals.

Methods| RMSE|  RMSCE|  NLL|  Check|Interval
-|-|-|-|-|-
 GaussianNLL| 1.504| **0.214**| **1.634**| **0.363**|**3.692**
-boot var|  |  0.570|  10689.836|  0.532|10.788
Quantile Regression|  0.015|  **0.253**|  NA|  **0.002**|0.219
-boot var | |  0.409 |  22.338 |  0.004 | **0.070**
Residual Prediction| 0.004 | **0.286**| NA| 0.005|0.061
-boot var| | 0.500 | 67.475| **0.001**|**0.025**

RMSE is an accuracy metric, which shows that applying GaussianNLL will severely influence the performance of the model, so the UQ of the model may not be considered valid. In the other models, the estimated variance seems to be more suited with proper scoring rules like the Interval score than interpolating the quantiles.

[1] Image-to-Image Regression with Distribution-Free Uncertainty Quantification and Applications in Imaging

[2] Uncertainty Toolbox: an Open-Source Library for Assessing, Visualizing,
and Improving Uncertainty Quantification

---

### Author Response · Authors · 2023-11-22

We extend our appreciations to all the reviewers for the impressive experience. Considering the current scores, it's a pity that ICLR may not publish it, yet I believe our work is meaningful in exploring new frontiers of the landscape.

---

### Meta-Review · Area_Chair_jWEp · 2023-12-06

**Metareview:**

This paper presents a method for MRI reconstruction. The authors claimed that current self-supervised MRI reconstruction methods rely on capturing the relationships between different views and the designs are often heuristic without deep insights. This work aims to address these issues and provides an explanation for self-supervised reconstruction.

However, the presentation of the paper needs significant improvement. The title seems to imply that the novelty lies on uncertainty quantification with a wide application in reconstruction. But it is actually for MRI reconstruction and the uncertainty quantification is not the claimed novelty.

**Justification For Why Not Higher Score:**

The presentation of the paper needs significant improvement for possible consideration of acceptance.

**Justification For Why Not Lower Score:**

NA

---

### Decision · Program_Chairs · 2024-01-16

Reject